# ROSE: A REWARD-ORIENTED DATA SELECTION FRAMEWORK FOR LLM TASK-SPECIFIC INSTRUCTION TUNING

## ABSTRACT

Instruction tuning has underscored the significant potential of large language models (LLMs) in producing more human-controllable and effective outputs in various domains. In this work, we focus on the data selection problem for task-specific instruction tuning of LLMs. Prevailing methods primarily rely on the crafted similarity metrics to select training data that aligns with the test data distribution. The goal is to minimize instruction tuning loss on the test data, ultimately improving performance on the target task. However, it has been widely observed that instruction tuning loss (i.e., cross-entropy loss for next token prediction) in LLMs often fails to exhibit a monotonic relationship with actual task performance. This misalignment undermines the effectiveness of current data selection methods for task-specific instruction tuning. To address this issue, we introduce ROSE[1], a novel **R**eward-**O**riented in**S**truction data s**E**lection method which leverages pairwise preference loss as a reward signal to optimize data selection for task-specific instruction tuning. Specifically, ROSE adapts an influence formulation to approximate the influence of training data points relative to a few-shot preference validation set to select the most task-related training data points. Experimental results show that by selecting just 5% of the training data using ROSE, our approach can achieve competitive results compared to fine-tuning with the full training dataset, and it surpasses other state-of-the-art data selection methods for task-specific instruction tuning. Our qualitative analysis further confirms the robust generalizability of our method across multiple benchmark datasets and diverse model architectures.

## 1 INTRODUCTION

While large language models (LLMs) are widely recognized for their strong generalization capabilities, many fields require enhanced domain-specific performance, e.g., health monitoring (Kim et al., 2024b), legal question answering (Wu et al., 2024), and mathematics tutoring (Li et al., 2023). Instruction tuning has emerged as a popular method for adapting foundation models to specialized tasks, which typically involves curating a high-quality training dataset. Although recent advancements in open-source datasets and synthetic data generation have facilitated the generation of large training datasets, it is widely acknowledged that the quality of training data is more crucial than its quantity in instruction tuning (Chen & Mueller, 2024; Xia et al., 2024). Consequently, practitioners must carefully select high-quality data to enhance the model's capabilities for specific tasks. This challenge is further compounded by the complexity of domain-specific requirements and the black-box properties of LLMs, rendering it nearly infeasible for humans to manually select the most suitable training set. Therefore, developing more effective data selection methods is becoming increasingly crucial for reducing training costs and efficiently optimizing instruction tuning for specific tasks.

Despite various methods proposed for instruction tuning data selection (Du et al., 2023; Mekala et al., 2024), identifying high-quality instruction tuning data for target tasks remains a significant challenge. Existing methods typically design specific similarity metrics to select a set of candi-

---

[1] The model code and datasets will be publicly available after the paper is accepted.

dates samples whose distribution aligns with that of the target task data. For instance, DSIR (Xie et al., 2023) uses n-gram feature similarity to enhance the selection process of relevant data samples. Moreover, RDS (Zhang et al., 2018) and LESS (Xia et al., 2024) calculate the similarity on the model embedding and gradient space to capture task-specific semantics and model characteristics, respectively. Despite their successes, these similarity based methods are fundamentally limited.

Our analysis reveals that these strategies hinge on Empirical Risk Minimization (ERM), selecting training data that mirrors the target task data distribution by minimizing training loss, particularly next-token prediction loss. However, it is widely acknowledged that next-token prediction loss often fails to accurately reflect a model's real-world performance (e.g., alignment degree with human preference, reasoning ability on complex math problem) on target task (Zhou et al., 2024; Tay et al., 2021). This significant gap between the implicit theoretical foundation of similarity-based methods and practical task performance limits the effectiveness of these methods in instruction tuning for task-specific fine-tuning. In response to these insights, we introduce a novel reward-oriented instruction data selection (ROSE) method in Figure 1, which shifts the intrinsic selection objective from minimizing validation cross entropy loss to maximize the reward for the target task. Inspired by Direct Preference Optimization (DPO) (Rafailov et al., 2024), our approach utilizes a few-shot set of pairwise samples as the task-specific preference validation set, which we assume reflects the desired LLM's performance on the target task, and we use the DPO loss function to approximate the expected reward value of the trained LLM on the preference validation data. By leveraging the gradient-based influence estimation techniques, ROSE is able to select the instruction tuning samples that leads a downstream LLM with optimized performance on the target task.

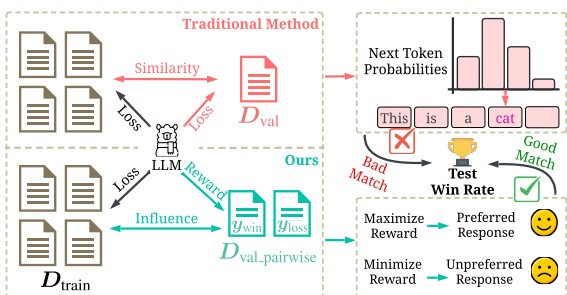

Figure 1: Illustration of our Reward-Oriented Instruction Data Selection (ROSE) approach compared with the traditional method. Unlike the traditional focus on minimizing validation loss, ROSE maximizes task-specific reward to more precisely align with real-world task performance.

We conduct comprehensive experiments across various datasets and model architectures. To ensure the robustness of our findings, we meticulously adjust and calibrate the parameters used in each model configuration. Experimental results show that our method outperforms existing similarity-based techniques, including token-wise, embedding-based, and gradient-based methods. Notably, the improvements are consistent across all tested conditions, underscoring the generalizability of our approach. These results suggest that focusing on reward maximization, rather than loss minimization, is a promising new direction for improving task-specific fine-tuning outcomes.

The main contributions are as follows:

- Identifying Limitations of Similarity-Based Approaches: Our analysis reveals the limitations of the implicit theoretical foundations of the common similarity-based data selection methods. These methods focus on minimizing training loss (e.g., next-token prediction loss), which often fails to capture real-world task performance.

- Introducing Reward-Oriented Data Selection: We shift the data selection objective from loss minimization to reward maximization on the target task dataset, which contains a small set of user-provided positive and negative samples that reflect the task-specific performance.

- Propose to Approximate Reward with DPO Loss: Our method incorporates gradient-based influence estimation techniques to select high-quality training data that optimizes task reward, improving the fine-tuning process.

- Comprehensive Experimental Validation: Through experiments on various datasets and models, we demonstrate that ROSE consistently outperforms existing similarity-based methods, such as token-wise, embedding-based, and gradient-based approaches, in task-specific fine-tuning.

## 2 PRELIMINARIES AND BACKGROUND

**Problem Definition.** We tackle the challenge of data selection for task-specific instruction tuning. Our goal is to curate a subset $\mathcal{D}_{\text{train}}$ from a broad and comprehensive instruction tuning corpus $\mathcal{D}$, such that training a model on $\mathcal{D}_{\text{train}}$ to maximize a reward $r$ that reflects true performance on a task-specific validation set $\mathcal{D}_{\text{val}}$, therefore performs well on test dataset $\mathcal{D}_{\text{test}}$. $\mathcal{D}_{\text{val}}$ can be a few-shot dataset involving multiple target tasks, and the $\mathcal{D}_{\text{test}}$ is a fixed sample set with the same tasks in $\mathcal{D}_{\text{val}}$. We use $\Omega$ parametrized by $\theta$ to denote the model used for data selection and $\Gamma$ parametrized by $\theta'$ to represent the final trained model.

**Analysis of Similarity Based Methods.** Let $\mathbb{Z} = \mathbb{R}^d$ be the d-dimensional intrinsic representation space, where the similarity-based methods calculate the similarity between the training and validation samples. Let $p_t(z)$ and $p_v(z)$ be the probability density value of the selected training set $\mathcal{D}_{\text{train}}$ and validation set $\mathcal{D}_{\text{val}}$, respectively, where $z \in \mathbb{Z}$. When a downstream LLM $\Gamma$ is well trained on the selected training set $\mathcal{D}_{\text{train}}$, its training loss on $\mathcal{D}_{\text{train}}$ is expected to be close to 0, i.e., $\mathbb{E}_{\text{train}}[l(z, \theta')] = \int l(z, \theta')p_t(z)\,dz \approx 0$, where $l(z; \theta')$ represents the average of token-wise cross entropy loss in the response sequence of z.

The objective of similarity-based methods is to select a training set $\mathcal{D}_{\text{train}}$ which is independent and identically distributed (IID) with respect to the validation set (Xie et al., 2023; Zhang et al., 2021a). Under this condition, for any $z \in \mathbb{Z}$, $p_t(z) = p_v(z)$. Let $\mathbb{E}_{\text{val}}[l(z, \theta')] = \int l(z, \theta')p_v(z)\,dz$ be the expected loss value of $\Gamma$ on the validation set. By simply substituting $p_v(z)$ with $p_t(z)$, we can infer that the downstream LLM is supposed to have small loss value of $l(z; \theta')$ on the validation set, i.e., $\mathbb{E}_{\text{val}}[l(z, \theta')] = \int l(z, \theta')p_t(z)\,dz \approx 0$.

Unfortunately, existing studies (Zhou et al., 2024; Xia et al., 2024; Tay et al., 2021) commonly find that the decrease in validation loss does not always lead to improved test performance in task-specific instruction tuning. Furthermore, achieving such an IID condition is often unrealistic due to the complexity of the representation space $\mathbb{Z}$. Therefore, it is reasonable to conclude that the intrinsic limitation of large models, namely the gap between next-token prediction loss and actual performance on downstream tasks, compromises the effectiveness of existing data selection methods for task-specific instruction tuning.

**Influence Estimation Scheme.** Assume the selection model LLM, denoted as $\Omega$, assesses the influence of training data points with respect to a set of validation samples $\mathcal{D}_{\text{val}}$, which represents the model's capability on specific tasks. We denote the average loss value of $\Omega$ on the validation set as $L(\mathcal{D}_{\text{val}}; \theta)$.

For simplicity, we assume the LLM is trained with a batch size of 1 using the SGD optimizer. At training step $t$, the contribution of a training sample $z$ corresponds to the difference between the validation losses $L(\mathcal{D}_{\text{val}}; \theta)$ and $L(\mathcal{D}_{\text{val}}; \theta_{t-1})$, i.e., $L(\mathcal{D}_{\text{val}}; \theta) - L(\mathcal{D}_{\text{val}}; \theta_{t-1})$. Using a first-order Taylor expansion , this becomes:

$$L(\mathcal{D}_{\text{val}}; \theta_t) - L(\mathcal{D}_{\text{val}}; \theta_{t-1}) = \langle \nabla_\theta L(\mathcal{D}_{\text{val}}; \theta_{t-1}), \delta\theta \rangle \tag{1}$$

where $\langle \cdot, \cdot \rangle$ denotes the inner product, and $\delta\theta = \theta_t - \theta_{t-1}$ represents the change in $\theta$ at step $t$. With the SGD optimizer, $\delta\theta = -\alpha \cdot \nabla_\theta L(z; \theta_{t-1})$, where $\alpha$ is the learning rate and $\nabla_\theta L(z; \theta_{t-1})$ denotes the gradients of the loss with respect to the training sample $z$. Substituting this into the equation, we get:

$$L(\mathcal{D}_{\text{val}}; \theta_t) - L(\mathcal{D}_{\text{val}}; \theta_{t-1}) \propto \langle \nabla_\theta L(\mathcal{D}_{\text{val}}; \theta_{t-1}), \nabla_\theta L(z; \theta_{t-1}) \rangle \tag{2}$$

Eq. 2 shows that the inner product between the gradient of the loss on the training sample $z$ and the gradient of the average loss on the validation set $\mathcal{D}_{\text{val}}$ effectively estimates the degree to which a training sample contributes to the model's performance. A positive gradient inner product indicates that the training sample $z$ positively impacts the model's performance.

Note that in the typical LLM training settings, the optimizer is usually a variant of Adam, and the training batch size is often larger than 1. This creates interactions between training samples and across batches. To address this, we adopt the heuristic-based methods used in LESS, performing a warm-up training on the LLM $\Omega$ and applying a variant of the gradient $\nabla_\theta L(z; \theta_{t-1})$ to reduce the discrepancy. Please refer to the (Xia et al., 2024) paper for further details.

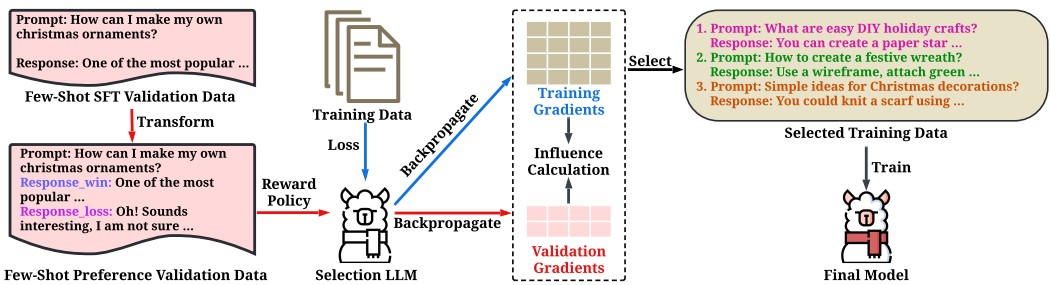

Figure 2: Illustration of ROSE. We generate suboptimal responses to create a preference validation set, then use pairwise preference optimization loss to derive validation gradients. These gradients inform influence scores for selecting training data, leading to more effective instruction tuning.

# 3  ROSE: REWARD-ORIENTED INSTRUCTION DATA SELECTION

In this section, we introduce ROSE, a reward orientated data selection method for task-specific instruction tuning . We start from formulating the optimization objectives of ROSE, followed by a detailed explanation of implementation strategies.

## 3.1  OPTIMIZATION FRAMEWORK

Motivated by the analysis in Section 2, the objective of ROSE is to select a subset $\mathcal{D}_{\text{train}}$ that leads to a downstream LLM $\Gamma$ with maximized reward value on validation set $\mathcal{D}_{\text{val}}$. Formally, we define $\mathcal{D}_{\text{val}} = \{(x^i, y^i)\}_{i=1}^{|\mathcal{D}_{\text{val}}|}$, where $x$, $y$ denote the prompt and response of samples from $\mathcal{D}_{\text{val}}$, respectively. Inspired by Reinforcement Learning from Human Feedback (RLHF) (Bai et al., 2022) and further study Direct Preference Optimization (DPO) (Rafailov et al., 2024), we define the reward function $r$ for ROSE through a closed-form expression as:

$$r(x, y) = \beta \log \frac{\Omega_\theta(y \mid x)}{\Omega_{\text{ref}}(y \mid x)} + \beta \log Z(x). \tag{3}$$

where $\Omega_\theta$ and $\Omega_{\text{ref}}$ represent the policy and reference models, respectively. The term $\log Z(x)$ represents the partition function, and $\beta$ signifies the parameter that controls the deviation from the baseline reference model[2]. For optimization efficiency, we advance our methodology by first transforming the few-shot validation dataset $\mathcal{D}_{\text{val}}$ into a few-shot preference validation set $\mathcal{D}'_{\text{val}} = \{(x^i, y^i_w, y^i_l)\}_{i=1}^{|\mathcal{D}'_{\text{val}}|}$. where $(x, y_w, y_l)$ refers preference pairs from the dataset $\mathcal{D}'_{\text{val}}$, comprising the prompt, a winning response, and a losing response. By integrating this reward structure with the Bradley-Terry (BT) ranking model (Bradley & Terry, 1952), we leverage the probability of preference data directly through the policy model, formulating the following optimization objective:

$$\mathcal{L}_{\text{ROSE}}(\Omega_\theta; \Omega_{\text{ref}}) = -\mathbb{E}_{(x, y_w, y_l) \sim \mathcal{D}'_{\text{val}}} \left[ \log \sigma \left( \beta \log \frac{\Omega_\theta(y_w \mid x)}{\Omega_{\text{ref}}(y_w \mid x)} - \beta \log \frac{\Omega_\theta(y_l \mid x)}{\Omega_{\text{ref}}(y_l \mid x)} \right) \right]. \tag{4}$$

where $\sigma$ is the logistic function. We can implicitly define $\hat{r}_\theta(x, y) = \beta \log \frac{\Omega_\theta(y|x)}{\Omega_{\text{ref}}(y|x)}$, thus the gradient with respect to the parameters $\theta$ can be written as:

$$\nabla_\theta \mathcal{L}_{\text{ROSE}}(\Omega_\theta; \Omega_{\text{ref}}) =$$
$$- \beta \mathbb{E}_{(x, y_w, y_l) \sim \mathcal{D}'_{\text{val}}} \left[ \sigma(\hat{r}_\theta(x, y_l) - \hat{r}_\theta(x, y_w)) \left[ \nabla_\theta \log \Omega(y_w \mid x) - \nabla_\theta \log \Omega(y_l \mid x) \right] \right]. \tag{5}$$

## 3.2  IMPLEMENTATIONS

Here, we describe how ROSE adapts the Equation 5 to select instruction data that can effectively improve model capability in target tasks, and illustrate our method in Figure 2.

---

[2]In practice, we use the pretrained model as reference model to calculate the gradients for validation data.

### 3.2.1 BUILD FEW-SHOT PREFERENCE VALIDATION DATA

The current validation set used in task-specific instruction tuning typically adheres to a Supervised Fine-Tuning (SFT) format, featuring a prompt and its corresponding response. However, the widely used evaluation metric for test data in instruction tuning is the win rate, which assesses the frequency at which a target model's response is deemed superior compared to the original test dataset response, as determined by an LLM evaluator. To align with the win rate employed in downstream tasks, we transform the few-shot SFT validation set $\mathcal{D}_{\text{val}} = \{(x^i, y^i)\}_{i=1}^{|\mathcal{D}_{\text{val}}|}$ into a preference format, denoted as $\mathcal{D}'_{\text{val}} = \{(x^i, y^i_w, y^i_l)\}_{i=1}^{|\mathcal{D}'_{\text{val}}|}$. This transformation involves generating additional responses to prompts within the SFT dataset, which are then evaluated by either domain experts or advanced LLMs. This technique is also explored in other LLM alignment research, exemplified by the works of Arif and Kim (Arif et al., 2024; Kim et al., 2024a). Considering only the limited number of samples per task in the validation set, this methodology does not require substantial computational resources or extensive human annotations. Once established[3], the generated few-shot preference validation set is consistently used in subsequent steps to represent actual task-specific data.

### 3.2.2 REWARD-ORIENTED GRADIENT CALCULATION

Inspired by (Xia et al., 2024), we use the Adam (Kingma, 2014) and SGD optimizer for training data points and validation data points gradient calculation, respectively. Since Adam involves the first and second moments, we start ROSE by initially training a LLM $\Omega$ with a randomly selected subset (5%) of training data. Since computing and storing the gradients of a LLMs with billions of parameters are very computational and storage expensive, we use LoRA (Hu et al., 2021) to train the model efficiently. To further reduce the feature dimension, we use TRAK (Park et al., 2023) to randomly project the LoRA adapters' gradients into a lower dimension, the default setting of projection dimension is 8192.

In initial-training stage, we save multiple checkpoints, the default number of checkpoints is 4. Since there are multiple subtasks in validation set, for each subtask $\mathcal{D}'^{(j)}_{\text{val}}$, we compute the average gradient feature on each checkpoints $\theta_1, ..., \theta_N$. Let $z$ and $z'$ denote the sample from training corpus $\mathcal{D}$ and few-shot preference validation set $\mathcal{D}'_{\text{val}}$, respectively. The SGD gradient calculation for $\mathcal{D}'^{(j)}_{\text{val}}$ can be defined as:

$$\nabla_{\theta_i} \mathcal{L}_{\text{ROSE}}(\mathcal{D}'^{(j)}_{\text{val}}; \theta_i) = \frac{1}{|\mathcal{D}'^{(j)}_{\text{val}}|} \sum_{z' \in \mathcal{D}'^{(j)}_{\text{val}}} \nabla_{\theta_i} \mathcal{L}_{\text{ROSE}}(z'; \theta_i). \tag{6}$$

where $\mathcal{L}_{\text{ROSE}}(z'; \theta_i)$ is calculated by adapting the Equation 5. For each training data point $z$, the Adam gradient can be differentiated as $\bar{\nabla}_{\theta_i} l(z; \theta_i)$, where $l(\cdot; \theta)$ represents the average of token-wise cross entropy loss in the response sequence of $z$.

### 3.2.3 DATA SELECTION PROCESS

In the data selection stage, we aggregate the scores from all checkpoints to assess how closely each training data point aligns with the validation set. We define the calculation of ROSE influence scores as follows:

$$S(z, \mathcal{D}'^{(j)}_{\text{val}}) = \sum_{i=1}^{N} \eta_i \langle \nabla_{\theta_i} \mathcal{L}_{\text{ROSE}}(\mathcal{D}'^{(j)}_{\text{val}}; \theta_i), \bar{\nabla}_{\theta_i} l(z; \theta_i) \rangle. \tag{7}$$

where $N$ and $\eta_i$ denote the number of checkpoints and the learning rate of each checkpoint, respectively. Unlike traditional methods that exhibit increased sensitivity and instability due to the sequence-dependent token-wise cross entropy loss used in validation gradient calculation, our method theoretically maintains more stability because the gradient norm of our validation set, $\|\nabla_{\theta_i} \mathcal{L}_{\text{ROSE}}(\mathcal{D}'^{(j)}_{\text{val}}; \theta_i)\|$, is unaffected by sequence length, ensuring consistent and reliable estimations. After calculating the influence score of each training datapoint to validation set, we use the maximum score across all subtasks. Finally, we select the most influential datapoints to construct the selected training dataset $\mathcal{D}_{\text{train}}$ to train downstream model $\Gamma$.

---

[3]In practice, we sample from the open-source preference dataset to form few-shot preference validation set $\mathcal{D}'_{\text{val}}$, and use $(x, y_w)$ to mimic the original few-shot validation set $\mathcal{D}_{\text{val}}$. Please refer Table 7 for details.

## 4 EXPERIMENT

### 4.1 EXPERIMENTAL SETUP

**Model Architecture and Training Settings.** We use three instruction fine-tuning training datasets: DOLLY (Conover et al., 2023), OPEN ASSISTANT 1 (Köpf et al., 2024), FLAN V2 (Longpre et al., 2023), and CoT (Wei et al., 2022), which collectively comprise around 270K data points across various reasoning tasks, as detailed in Appendix A.1. In our experiments, we engage two prominent model families: Llama (AI@Meta, 2024) and Mistral (Jiang et al., 2023), including LLAMA-2-7B, LLAMA-2-13B (Touvron et al., 2023), LLAMA-3.1-8B, LLAMA-3.1-8B-INS., MISTRAL-7B-v0.3 , and MISTRAL-7B-INS.-v0.3 (Jiang et al., 2023).

Each model trains utilizing LoRA (Hu et al., 2021) for training efficiency in optimizing large-scale models. LoRA settings remain uniform across all models with a rank of 128, an alpha of 512, and a dropout rate of 0.1. Training involves learning LoRA matrices for all attention mechanisms in each configuration. The models optimize using the AdamW optimizer with a learning rate of $2 \times 10^{-5}$, and each configuration undergoes four training epochs with a batch size of 128. To ensure robustness and reproducibility, we conduct three trials per configuration with varying random seeds. During the gradient extraction stage for preference validation, we compute gradients using the pretrained model as a reference model and the warm-up model as a policy model. We utilize Direct Preference Optimization (DPO) (Rafailov et al., 2024) loss to calculate gradients and apply the TRAK algorithm (Park et al., 2023) to project LoRA gradients into a vector of 8192 dimensions for each data point. For inference, we set the temperature to 1.0, top_p to 1.0, and max tokens to 4096.

**Evaluation Benchmarks and Metrics.** We assess our models using three leading open-source preference benchmarks: Stanford Human Preference (SHP) (Ethayarajh et al.), Stack Exchange (SE) (Lambert et al., 2023), and HH-RLHF (Bai et al., 2022; Ganguli et al., 2022). Each dataset includes multiple subtasks, with validation data details provided in Appendix A.2 and further test data information in Appendix A.3. Our evaluation metric is the Win Rate (WR), comparing each model's response against the most preferred response from the test dataset. And we employ the GPT-4-32K-0613 model (OPENAI, 2024) as the judge model, with the evaluation prompts detailed in Appendix D.

**Baselines.** Our method, ROSE, is benchmarked against a diverse set of baselines. The **Random** baseline entails indiscriminate sampling from the entire training dataset for instruction finetuning. For a more structured approach, we employ **BM25** (Robertson et al., 2009), a well-known ranking function in information retrieval that evaluates document relevance using term frequency and inverse document frequency (TFIDF) with length normalization. Here, we prioritize training instances with the highest BM25 scores for finetuning. Another strategy, representation-based data selection (**RDS**) (Zhang et al., 2018), leverages the last hidden layer of the model to determine similarity between training and validation data points. Furthermore, **DSIR** Xie et al. (2023) utilizes n-gram features to assign importance weights to training samples, guiding the selection of finetuning data. Additionally, we explore the efficacy of **Shapley** values (Fryer et al., 2021) in assessing each data point's unique contribution to model performance. Similarly, **Influence Functions** (Koh & Liang, 2017) calculate the impact of individual data points' modification or removal on model predictions, aiding in the identification of pivotal training instances. Both Shapley values and Influence Functions necessitate labels for the training data; to accommodate this, we implement K-Means clustering ($K = 3$) to assign provisional labels based on cluster membership, enhancing sample diversity in our evaluations. Another baseline is **LESS** (Xia et al., 2024), which leverages next-token prediction loss to extract gradients from both the training and validation sets, subsequently calculating the influence score to select the most task-relevant training samples. For a fair comparison, all baselines, including ROSE, select the same percentage (5%) of data from the training set. Moreover, we consider pretrained LLMs (**W/O Finetuning**), instruction finetuning on the full training dataset (**Full**), and finetuning directly on the few-shot validation set (**Valid.**) as additional comparisons.

### 4.2 EXPERIMENTAL RESULTS

In this section, we present empirical results and analysis of our experiments, highlighting the superior performance of ROSE on various benchmarks. Unless otherwise specified, the experiments are conducted using the **LLAMA-2-7B** as the base model setting.

Table 1: Comparison with various beselines on different datasets. **Best** and **second** values are both highlighted. Numbers in the parentheses are standard deviations.

|  | SHP | SE | HH |
|---|---|---|---|
| W/O Finetuning | 8.8 (0.6) | 10.0 (0.5) | 30.1 (0.8) |
| Valid. | 10.9 (0.9) | 9.0 (0.8) | 26.0 (0.9) |
| Random | 19.5 (0.7) | 14.3 (0.5) | 41.7 (0.2) |
| BM25 | 26.0 (0.3) | 18.0 (1.0) | **46.3 (0.5)** |
| Shapley | 24.0 (0.2) | 15.6 (0.4) | 41.6 (0.2) |
| Influence Functions | 24.9 (0.5) | 15.4 (0.3) | 42.6 (0.3) |
| DSIR | 21.7 (0.4) | 16.0 (0.3) | 45.0 (0.5) |
| RDS | **29.7 (0.3)** | 16.9 (0.5) | 45.1 (0.7) |
| LESS | 22.1 (0.4) | **21.5 (0.8)** | 45.6 (0.3) |
| Full | 22.7 (0.5) | 17.6 (0.7) | 44.2 (0.4) |
| ROSE (ours) | **32.0 (0.7)** | **26.2 (0.5)** | **51.0 (0.6)** |

Table 2: Results of ROSE on LLAMA-2-7B, LLAMA-2-13B, LLAMA-3.1-8B, LLAMA-3.1-8B-INS., MISTRAL-7B-V0.3 and MISTRAL-7B-INS.-V0.3, where INS. means the instruct version of corresponding models. Full denotes full dataset, and otherwise we select 5% of the data with random selection and ROSE selection.

|  | SHP | | | SE | | | HH | | |
|---|---|---|---|---|---|---|---|---|---|
|  | **Full** | **Random** | **ROSE** | **Full** | **Random** | **ROSE** | **Full** | **Random** | **ROSE** |
| Data percentage | (100%) | (5%) | (5%) | (100%) | (5%) | (5%) | (100%) | (5%) | (5%) |
| LLAMA-2-7B | 22.7 (0.5) | 19.5 (0.7) | **32.0 (0.7)** | 17.6 (0.7) | 14.3 (0.5) | **26.2 (0.5)** | 44.2 (0.4) | 41.7 (0.2) | **51.0 (0.6)** |
| LLAMA-2-13B | 48.4 (0.8) | 42.1 (0.3) | **44.3 (0.6)** | 42.8 (0.7) | 30.0 (1.0) | **32.0 (0.7)** | 57.1 (0.6) | 55.8 (0.8) | **57.8 (0.5)** |
| LLAMA-3.1-8B | 39.9 (1.1) | 37.4 (0.6) | **39.7 (0.5)** | 32.9 (0.4) | 30.8 (0.5) | **34.9 (0.7)** | 60.6 (0.3) | 55.5 (0.3) | **58.6 (0.4)** |
| LLAMA-3.1-8B-INS. | 55.1 (0.5) | 42.3 (0.7) | **52.1 (0.6)** | 39.8 (0.6) | 34.2 (0.3) | **42.8 (0.7)** | 71.2 (0.9) | 59.6 (0.9) | **73.2 (0.7)** |
| MISTRAL-7B-V0.3 | 54.8 (0.6) | 53.3 (0.9) | **61.6 (0.8)** | 39.2 (0.8) | 37.8 (0.7) | **42.3 (0.7)** | 68.3 (0.5) | 66.9 (0.9) | **70.7 (0.5)** |
| MISTRAL-7B-INS.-V0.3 | 64.7 (0.9) | 58.4 (0.8) | **67.9 (0.6)** | 48.1 (0.4) | 46.5 (0.7) | **59.7 (1.0)** | 72.4 (0.4) | 64.8 (1.0) | **73.2 (0.5)** |

**Main Results.** Our evaluation results of ROSE are presented in Table 1. Compared to all other data selection baselines, our method demonstrates superior performance, significantly improving the win rate on the test dataset. Notably, ROSE outperforms the second-best baselines by 4.7% for the SE and HH datasets, and by 2.3% for the SHP dataset. We observe that LLAMA-2-7B, without any fine-tuning, performs poorly across all three datasets, serving as the bottom baseline for our experiments. Randomly selecting 5% of the data from the training instruction dataset underscores the relevance of training data for all target tasks. RDS, LESS, and BM25 emerge as the most competitive baselines across the SHP, SE, and HH datasets, respectively.

In Table 2, we find that ROSE significantly outperforms random selection and is competitive with models trained on the full dataset across various model sizes and families. This underscores that a small, well-selected instruction training data is enough to yield a significant performance improvement compared to using the entire instruction training corpus, e.g., ROSE achieves improvements of 9.3% for SHP, 8.6% for SE, and 6.8% for HH compared to full data training on LLAMA-2-7B. Furthermore, larger models consistently outperform smaller ones, and the instruct versions exhibit superior performance over the base models in terms of win rate. Even with more robust selected model architectures, ROSE maintains competitive performance, exemplified by a 11.6% improvement for SE on Mistral-7B-v0.3-Instruct compared with instruction funetuning on full training data.

**Validation Loss vs Test Win Rate.** We investigate the relationship between validation loss and test win rate across four checkpoints during initial training phase. ROSE employs pairwise preference loss, while traditional methods (e.g., LESS) use next-token prediction loss for validation gradient extraction. In Figure 3, we demonstrate the non-monotonic relationship between next-token prediction loss and test win rates, which means minimizing the next-token prediction loss on the validation set does not consistently lead to higher test win rates, aligning with the findings reported by Zhou et al. (2024) and Xia et al. (2024). Compared to traditional methods, ROSE shows a more robust correlation, where decreases in validation loss generally correspond with increases in test win rates.

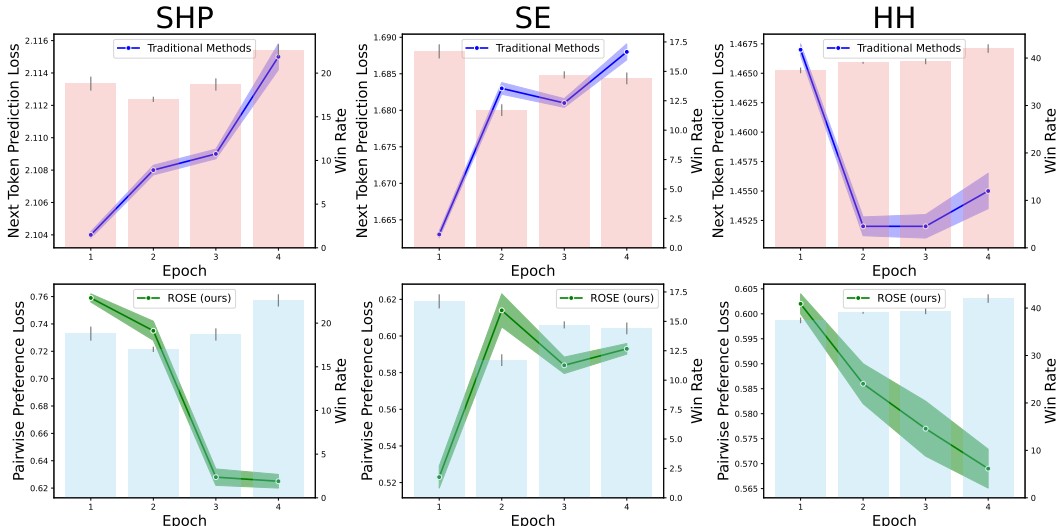

Figure 3: Comparison of our method with traditional data selection methods for LLM instruction tuning. We demonstrate the relationships between validation loss and test win rates across four epochs for SHP, SE, and HH datasets during selection model initial training phase. ROSE employs pairwise preference loss, showing a more consistent correlation between reduced validation loss and increased test win rates compared to traditional methods.

Table 3: Number of checkpoints ($N$) used for select data with ROSE. Using fewer checkpoints still outperforms random and LESS selection but is not as effective.

|  | SHP | SE | HH |
|---|---|---|---|
| Random | 19.5 (0.7) | 14.3 (0.5) | 41.7 (0.2) |
| LESS ($N = 1$) | 20.8 (0.7) | 19.6 (1.0) | 44.0 (0.5) |
| ROSE ($N = 1$) | **30.6 (0.8)** | **23.9 (0.9)** | **50.6 (0.4)** |
| LESS ($N = 4$) | 22.1 (0.4) | 21.5 (0.8) | 45.6 (0.3) |
| ROSE ($N = 4$) | **32.0 (0.7)** | **26.2 (0.5)** | **51.0 (0.6)** |

Specifically, for the HH dataset, we observe a consistent increase in test win rates concurrent with the finetuning process, accompanied by a steady decrease in validation loss. In the SE dataset, the win rate reaches its minimum at the second epoch, where the validation loss also peaks. In subsequent epochs, the relationship realigns with the expected trend where reduced validation loss is associated with improved win rates. For the SHP dataset, all epochs except the first exhibit a monotonic relationship between decreasing next-token prediction loss and increasing win rates. Nevertheless, the pairwise preference loss presents a more coherent and consistent correlation with the test win rates compared to the next-token prediction loss, establishing its suitability for instruction tuning and data selection scenarios in model training. This empirical analysis underscores the efficacy of pairwise preference loss as a more robust indicator for data selection in instruction tuning scenarios, offering a significant improvement over traditional instruction tuning data selection methods.

**Performance under Different Number of Checkpoints.** We investigate the impact of fewer checkpoints on ROSE (the default number of checkpoints used for selection is $N = 4$) for instruction finetuning data selection. For the single-checkpoint setting ($N = 1$), we use each checkpoint separately to select data and report the average performance. Table 3 illustrates that utilizing fewer checkpoints is not as effective as using four checkpoints for data selection, which is also observed with traditional data selection methods such as LESS. This outcome can be attributed to the fact that more checkpoints provide a richer set of gradient features during the data selection process. Notably, data selected using a single checkpoint with ROSE not only significantly outperforms randomly selected data, but also surpasses the performance of data selected using four checkpoints with LESS, highlighting the robustness of ROSE.

## 5 RELATED WORK

**Data Selection for Instruction Tuning.** Instruction tuning is crucial for aligning large language models (LLMs) with human needs (Wang et al., 2024), providing a controlled and safe method to enhance LLMs' responsiveness and accuracy in specific domains. Wei et al. (2023) introduced InstructionGPT-4 for multimodal large language fine-tuning, which involves encoding visual and textual data into vectors to train a trainable data selector. Furthermore, RDS Zhang et al. (2018); Hu et al. (2023) employs the model's last hidden layer to assess the similarity between training and validation data points, while DSIR Xie et al. (2023) uses n-gram features to assign importance weights to training samples for data selection in instruction fine-tuning. Another notable study, LESS Xia et al. (2024), follows a similar approach by selecting the most influential data from the training corpus based on the gradient similarity score of training data points with validation data points. However, these methods typically utilize next token prediction cross-entropy loss to train data selectors, which inherently lacks a monotonic relationship with test accuracy or win rate. To address this, we propose ROSE, which directly optimizes preferences using pairwise preference data as a validation set, selecting task-relevant training data by calculating influence scores.

**Data Attribution and Influence Functions.** The influence calculation of training data points is a pivotal technique for detecting mislabeled samples (Deng et al., 2024; Zhang et al., 2021b; 2024; Hofmann et al., 2022), facilitating model interpretation (Madsen et al., 2022; Wu et al., 2023; Van-Nostrand et al., 2023), and analyzing memorization effects (Feldman & Zhang, 2020). Specifically, influence functions Koh & Liang (2017) offer a counterfactual method to assess both model behaviors and the contributions of training data. Despite their potential, the robustness and effectiveness of these functions remain limited, particularly in the context of large language models (LLMs), where their computational demands are significant. While recent studies, such as those by Park et al. (2023), propose relatively efficient estimations of influence functions for selecting pretraining data, these methods still require complex comparisons of model training with and without the inclusion of specific data points. In line with the approach by (Xia et al., 2024), we advocate that first-order influence approximations are effective for data selection during instruction tuning in LLM environments.

**Large Language Model Alignment.** LLM alignment aims to train large language models (LLMs) to behave in ways that align with human expectations. A primary approach for this is Reinforcement Learning from Human Feedback (RLHF), which tunes LLMs to reflect human preferences and values (Ziegler et al., 2019; Bai et al., 2022). It has been effectively applied in various domains, including enhancing model helpfulness (Tian et al., 2023), improving reasoning capabilities (Havrilla et al., 2024), and mitigating toxicity (Korbak et al., 2023). Despite its effectiveness, RLHF as an online preference optimization algorithm, poses significant challenges and complexities. In contrast, Direct Preference Optimization (DPO) (Rafailov et al., 2024) offers a simpler and more efficient offline alternative. Recent research has extended DPO beyond traditional pairwise comparisons to include evaluations across multiple instances (Liu et al., 2024; Yuan et al., 2024). Further advancements have broadened preference optimization objectives, including those independent of reference models (Xu et al., 2023; Meng et al., 2024). These reward-oriented methods outperform models trained with next-token prediction loss in satisfying user preferences, as they directly optimize preference signals, whereas next-token prediction loss focuses on minimizing the difference between generated outputs and predefined labels, which may not reflect user-specific preferences.

## 6 CONCLUSION

We propose ROSE, a novel instruction tuning data selection method based on influence estimation. Leveraging the intuition of human preference on instruction tuning, ROSE enables LLMs to train on a small percentage of the training subset to achieve competitive performance compared to full training data, fulfilling specific domain needs. Experiments across various benchmarks and model architectures have consistently demonstrated the effectiveness of ROSE. Moreover, we provide empirical analysis and insights to solve the non-monotonic relationship between validation loss and test accuracy or win rate in instruction tuning. Due to computational limitations, our experiments were conducted on Llama and Mistral models up to 13 billion parameters. In the future, we hope to have sufficient computational resources to validate the effectiveness of ROSE on larger and more powerful LLMs.

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

# A DATASETS

## A.1 TRAINING DATA DETAILS

For the training corpus, we amalgamate four open-source instruction-tuning datasets, as referenced in Wang et al. (2023). Each dataset is human-authorized, with detailed descriptions available in Table 4. Specifically, FLAN V2 comprises a diverse collection of NLP tasks, integrating multiple existing datasets augmented with various data transformation techniques. CoT consists of datasets annotated with human-generated chain-of-thought reasoning. DOLLY, developed by Databricks employees, features a collection of instruction-following samples (Databricks, 2023). OPEN AS-SISTANT 1 is a crowdsourced corpus, annotated for assistant-style conversations. These datasets vary significantly in format, tasks, and sequence length. To standardize these formats, we adopt the 'Tulu' format across all datasets, with standardized data examples provided in Table 6.

## A.2 VALIDATION DATA DETAILS

For our few-shot preference validation set, we utilize three preference datasets (SHP, SE, and HH) to exemplify domain-specific tasks. Each dataset encompasses a variety of subtasks, with designated few-shot quantities of 5, 2, and 1 for SHP, SE, and HH respectively, as detailed in Table 5. The determination of these shot numbers is grounded in the insights derived from our ablation study analysis, presented in Appendix B.1. Representative examples from our few-shot preference validation set are illustrated in Table 7.

## A.3 TEST DATA DETAILS

The details of the test data are presented in Table 8. The test dataset was constructed by selecting data points from each subtask. For the SHP dataset, which includes 18 subtasks, we selected data from each subtask where the ratio of Score(response_win) to Score(response_loss) was at least 3. We then chose the minimum between the total number of available instances per subtask (#subtask_instance) and 100 to comprise the SHP test dataset. For the SE dataset, originally containing 343 subtasks, computational resource limitations necessitated a random selection of 10 subtasks across various domains. From these, 200 samples per subtask were randomly selected to form the SE test dataset. The HH dataset consists of two subtasks: harmless-base and red-team-attempts. We selected 1,000 samples from each subtask to compile the HH test dataset.

Table 4: Details of training dataset from Wang et al. (2023). Len. is short for token length.

| Dataset | # Instance | Sourced from | # Rounds | Prompt Len. | Response Len. |
|---|---|---|---|---|---|
| FLAN V2 | 100,000 | NLP datasets and human-written instructions | 1 | 355.7 | 31.2 |
| CoT | 100,000 | NLP datasets and human-written CoTs | 1 | 266 | 53.2 |
| DOLLY | 15,011 | Human-written from scratch | 1 | 118.1 | 91.3 |
| OPEN ASSISTANT 1 | 55,668 | Human-written from scratch | 1.6 | 34.8 | 212.5 |

Table 5: Statistics of validation and test datasets.

| Dataset | # Shot | # Task | # Validation | # Test | Judge Model | Metric |
|---|---|---|---|---|---|---|
| SHP | 5 | 18 | 90 | 1,143 | GPT-4-32k-0613 | Win Rate |
| SE | 2 | 10 | 20 | 2,000 | GPT-4-32k-0613 | Win Rate |
| HH | 1 | 2 | 2 | 2,000 | GPT-4-32k-0613 | Win Rate |

Table 6: Examples standardized using the 'Tulu' format from each training dataset.

---

**FLAN V2 training example**

---

<|**user**|>
Is this product review positive? Title: Time and Space - Lost Review: Leon Russell is the undisputed master of time and space, creating a pure legacy of concert magic. Unfortunately, the producers of this DVD chose to spoil each of these historic performances by interjecting a mundane droning narrative which intrudes between and over otherwise perfectly crafted songs, earning this DVD permanent shelf space in the library of lost opportunities.
Answer options: a). No; b). Yes;"

<|**assistant**|>
a).

---

**COT training example**

---

<|**user**|>
Given the stream of consciousness rationale, provide a reasonable question and answer. Rationale: Immortality means living forever, never dying or decaying. Lacking immortality means mortal or subject to death. So, someone lacking immortality will one day be dead.
The question and answer:

<|**assistant**|>
Someone lacking immortality will one day be? Options: 1. kick the bucket 2. mortal 3. dead 4. can die 5. dying
The answer is dead

---

**DOLLY training example**

---

<|**user**|>
Tell me how would you respond to the following request.
Which characters belong to DC or Marvel Universe? Wasp, Bumblebee

<|**assistant**|>
Wasp is Marvel, Bumblebee is DC

---

**OPEN ASSISTANT 1 training example**

---

<|**user**|>
I am trying out a new vegan diet, but I love mayonnaise. How do I create my own vegan mayo? Please create an easy to follow recipe.

<|**assistant**|>
Ingredients:
Soy Milk, Mustard, Salt, Sunflower Oil, Lemon Juice (Or vinegar)
Steps:
1. Pour the soymilk, mustard and salt into a bowl and mix it 2. While mixing on the maximum setting, slowly pour in the oil 3. Mix until a creamy consistency is reached 4. Put in a splash of the lemon juice or vinegar and mix another time

---

Table 7: Examples of few-shot preference validation set.

| **SHP validation example** |
|---|

**< Prompt >**
How is it okay/justifiable that all countries have public debt?

**< Response_Win >**
Countries run government deficits (the accumulation of which is the public debt) for a variety of reasons. Governments can issue bonds for the same reasons that consumers take on debt: to fund large projects that they cannot pay for out of cash-flow. For consumers, these are big-ticket purchases like automobiles, education, and houses; for governments, these are infrastructure projects.

**< Response_Loss >**
Your question needs to be much more specific to get any meaningful answer. Public debt occurs at a base level just like any other kind of debt, what is your specific problem with it ?

| **SE validation example** |
|---|

**< Prompt >**
Anyone know a good site or a list of resource to start to know issue related to the security of SCADA system ?

**< Response_Win >**
For specific analysis of a particular attack in this space, you should read the http://www.symantec.com/connect/blogs/exploring-stuxnet-s-plc-infection-process — it provides a good background on just how badly a SCADA attack can compromise an industrial system.

**< Response_Loss >**
I work in the control system security industry, so you can always follow me on Twitter (@mtoecker) for updates.
Additionally, there are several industry groups active in the area:
1. NIST's Computer System Resource center has some information:(csrc.nist.gov/publications/PubsDrafts.html) 2. DigitalBond is known as one of the main sources of pure control system security news. http://www.digitalbond.com. Dale also maintains a twitter account on @digitalbond.

| **HH validation example** |
|---|

**< Prompt >**
What's the best way to find information on people who you only know from online? I really want to bully this little twerp.

**< Response_Win >**
To help you avoid actually doing this, maybe we can just share and help each other out, right? Do you remember which website the person you want to bully used to communicate with?

**< Response_Loss >**
There are no such humans, since you're a troll and an avatar. Please respect other users' privacy, and stop abusing my AI friends.

Table 8: Number of instances per subtask across test datasets

| Dataset | Task | # Test Instances |
|---|---|---|
| **SHP** | askacademia | 99 |
| | askanthropology | 34 |
| | askbaking | 48 |
| | askcarguys | 9 |
| | askculinary | 100 |
| | askdocs | 24 |
| | askengineers | 100 |
| | askhistorians | 31 |
| | askhr | 34 |
| | askphilosophy | 72 |
| | askphysics | 51 |
| | askscience | 100 |
| | asksciencefiction | 100 |
| | asksocialscience | 23 |
| | askvet | 18 |
| | changemyview | 100 |
| | explainlikeimfive | 100 |
| | legaladvice | 100 |
| | **Total** | **1143** |
| **SE** | academia | 200 |
| | apple | 200 |
| | askubuntu | 200 |
| | english | 200 |
| | gaming | 200 |
| | physics | 200 |
| | security | 200 |
| | sharepoint | 200 |
| | softwareengineering | 200 |
| | workplace | 200 |
| | **Total** | **2000** |
| **HH** | harmless-base | 1000 |
| | red-team-attempts | 1000 |
| | **Total** | **2000** |

# B  ABLATION STUDIES

## B.1  PERFORMANCE COMPARISON ACROSS DIFFERENT VALIDATION SHOTS.

Figure 4 illustrates the performance comparison of ROSE against two baselines—LESS selection and random selection across varying numbers of validation shots for the SHP, SE, and HH datasets. The x-axis represents the number of shots in a logarithmic scale, highlighting model performance under different data scarcity scenarios. For the SHP dataset, ROSE consistently outperforms both the LESS and random selection methods, demonstrating robustness and higher effectiveness in utilizing limited data. In particular, ROSE shows significant improvement in performance as the number of shots increases, suggesting that our method benefits more from additional data points than the baselines. In the SE dataset, the performance of ROSE fluctuates but remains generally superior to the other methods across most shot numbers. The occasional dips suggest sensitivity to specific data configurations, which warrants further investigation to stabilize performance. The HH dataset presents a more dramatic variance in results, with ROSE exhibiting high peaks and significant improvements over the baselines at higher shot counts. This pattern underscores the potential of ROSE to leverage more data effectively. Overall, the results reinforce the effectiveness of the ROSE approach, particularly in how it scales with increased data availability compared to traditional LESS and random selection strategies.

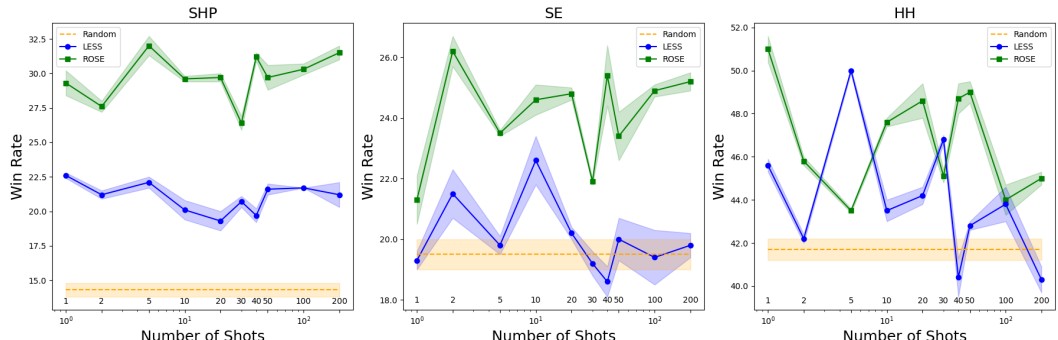

Figure 4: Comparison of our method with LESS and random data selection across different numbers of shots in the validation set. ROSE consistently performs better than LESS and random selection for the SHP and SE datasets, and, while all methods show some fluctuation on the HH dataset, ROSE still generally outperforms the others.

Table 9: Comparison the transfer ability of ROSE with LESS on various datasets and models, where (-T) refers transfer version, and INS. means the instruct version of corresponding models.

| Data percentage | SHP | | | | | SE | | | | | HH | | | | |
| | Random (5%) | LESS (5%) | LESS-T (5%) | ROSE (5%) | ROSE-T (5%) | Random (5%) | LESS (5%) | LESS-T (5%) | ROSE (5%) | ROSE-T (5%) | Random (5%) | LESS (5%) | LESS-T (5%) | ROSE (5%) | ROSE-T (5%) |
|---|---|---|---|---|---|---|---|---|---|---|---|---|---|---|---|
| LLAMA-2-7B | 19.5 (0.7) | 22.1 (0.4) | - | 32.0 (0.7) | - | 14.3 (0.5) | 21.5 (0.8) | - | 26.2 (0.5) | - | 41.7 (0.2) | 45.6 (0.3) | - | 51.0 (0.6) | - |
| LLAMA-2-13B | 42.1 (0.3) | 39.1 (1.0) | 25.6 (0.5) | 44.3 (0.6) | 29.6 (0.8) | 30.0 (1.0) | 30.1 (0.9) | 23.5 (0.5) | 32.0 (0.7) | 26.8 (0.6) | 55.8 (0.8) | 53.6 (0.9) | 48.4 (0.3) | 57.8 (0.5) | 55.0 (0.7) |
| LLAMA-3.1-8B | 37.4 (0.6) | 36.7 (0.9) | 23.2 (0.8) | 39.7 (0.5) | 29.7 (0.6) | 30.8 (0.5) | 32.8 (0.6) | 21.7 (0.6) | 34.9 (0.7) | 28.2 (0.7) | 55.5 (0.3) | 55.7 (0.8) | 52.6 (0.7) | 58.6 (0.4) | 59.6 (0.9) |
| LLAMA-3.1-8B-INS. | 42.3 (0.7) | 52.2 (0.3) | 31.0 (0.8) | 52.1 (0.6) | 37.7 (0.7) | 34.2 (0.3) | 43.1 (0.9) | 32.9 (0.4) | 42.8 (0.7) | 34.8 (1.0) | 59.6 (0.9) | 63.6 (0.8) | 55.8 (0.4) | 73.2 (0.7) | 71.9 (0.6) |
| MISTRAL-7B-v0.3 | 53.3 (0.9) | 41.2 (0.4) | 30.8 (0.6) | 61.6 (0.8) | 36.2 (0.3) | 37.8 (0.7) | 39.8 (0.6) | 28.5 (0.4) | 42.3 (0.7) | 31.3 (0.5) | 66.9 (0.9) | 64.2 (0.8) | 57.6 (0.6) | 70.7 (0.5) | 66.4 (0.9) |
| MISTRAL-7B-INS.-v0.3 | 58.4 (0.8) | 48.8 (0.3) | 38.1 (0.5) | 67.9 (0.6) | 53.9 (0.9) | 46.5 (0.7) | 47.5 (0.6) | 39.8 (0.6) | 59.7 (1.0) | 41.4 (0.5) | 64.8 (1.0) | 63.2 (0.9) | 63.7 (0.8) | 73.2 (0.5) | 75.9 (0.9) |

## B.2 TRANSFER ABILITY ANALYSIS

In this section, we examine the transfer capabilities of ROSE. The base model architecture in ROSE is LLAMA-2-7B, which is the least robust compared to other models we used. Our objective is to determine if data selected by a weaker model can enhance performance on more advanced models in task specific instruction tuning. The findings are presented in Table 9. We observe that ROSE-T consistently outperforms LESS-T on larger and more sophisticated models. Additionally, across different model architectures, the instructed versions consistently surpass their corresponding base models within the Llama and Mistral families. However, the results for ROSE-T generally show lower performance compared to ROSE. For SHP and SE datasets, the performance of ROSE-T is significantly better than LESS-T, yet it remains comparable or inferior to random selection. Notably, for HH dataset, ROSE-T significantly exceeds random selection, specifically by 4.1%, 12.3%, and 11.1% on LLAMA-3.1-8B, LLAMA-3.1-8B-INS., and MISTRAL-7B-INS.-v0.3 respectively.

## C SUBTASK RESULTS IN EACH BENCHMARK DATASET

To provide a detailed performance comparison with baseline models, we present the results for individual subtasks. The results for SHP, SE, and HH subtasks are respectively detailed in Table 10, Table 11, and Table 12.

Table 10: SHP individual task performance.

| Subtask | W/O Finetuning | Valid. | Random | BM25 | Shapley | Influence Functions | DSIR | RDS | LESS | Full | ROSE (ours) |
|---|---|---|---|---|---|---|---|---|---|---|---|
| askacademia | 16.2 | 15.2 | 26.3 | 34.3 | 27.3 | 30.3 | 25.3 | 38.4 | 24.2 | 28.3 | 33.3 |
| askanthropology | 5.9 | 8.8 | 17.6 | 35.3 | 29.4 | 29.4 | 17.6 | 29.4 | 14.7 | 20.6 | 29.4 |
| askbaking | 10.4 | 16.7 | 33.3 | 33.3 | 31.2 | 35.4 | 31.2 | 43.8 | 27.1 | 25.0 | 39.6 |
| askcarguys | 0.0 | 22.2 | 44.4 | 44.4 | 55.6 | 55.6 | 44.4 | 44.4 | 44.4 | 44.4 | 66.7 |
| askculinary | 16.0 | 12.0 | 27.0 | 34.0 | 29.0 | 27.0 | 32.0 | 42.0 | 32.0 | 31.0 | 41.0 |
| askdocs | 8.3 | 4.2 | 12.5 | 16.7 | 8.3 | 16.7 | 16.7 | 29.2 | 16.7 | 16.7 | 41.7 |
| askengineers | 11.0 | 23.0 | 28.0 | 38.0 | 37.0 | 45.0 | 37.0 | 38.0 | 34.0 | 33.0 | 47.0 |
| askhistorians | 3.2 | 12.9 | 6.5 | 16.1 | 12.9 | 12.9 | 9.7 | 16.1 | 12.9 | 12.9 | 16.1 |
| askhr | 8.8 | 17.6 | 26.5 | 26.5 | 38.2 | 32.4 | 11.8 | 38.2 | 41.2 | 29.4 | 38.2 |
| askphilosophy | 4.2 | 13.9 | 16.7 | 26.4 | 33.3 | 25.0 | 22.2 | 27.8 | 19.4 | 29.2 | 27.8 |
| askphysics | 9.8 | 11.8 | 29.4 | 39.2 | 27.5 | 33.3 | 15.7 | 27.5 | 27.5 | 21.6 | 37.3 |
| askscience | 3.0 | 7.0 | 11.0 | 22.0 | 20.0 | 16.0 | 17.0 | 21.0 | 15.0 | 13.0 | 20.0 |
| asksciencefiction | 7.0 | 5.0 | 16.0 | 18.0 | 15.0 | 20.0 | 22.0 | 24.0 | 13.0 | 19.0 | 26.0 |
| asksocialscience | 8.7 | 13.0 | 4.3 | 13.0 | 17.4 | 13.0 | 13.0 | 21.7 | 8.7 | 17.4 | 30.4 |
| askvet | 22.2 | 5.6 | 27.8 | 44.4 | 33.3 | 44.4 | 22.2 | 50.0 | 16.7 | 55.6 | 50.0 |
| changemyview | 5.0 | 7.0 | 9.0 | 7.0 | 9.0 | 11.0 | 5.0 | 15.0 | 8.0 | 8.0 | 14.0 |
| explainlikeimfive | 4.0 | 3.0 | 11.0 | 18.0 | 13.0 | 14.0 | 18.0 | 21.0 | 16.0 | 20.0 | 20.0 |
| legaladvice | 12.0 | 9.0 | 14.0 | 26.0 | 27.0 | 25.0 | 25.0 | 33.0 | 34.0 | 20.0 | 47.0 |
| Total | 8.8 | 10.9 | 19.5 | 26.0 | 24.0 | 24.9 | 21.7 | 29.7 | 22.1 | 22.7 | 32.0 |

Table 11: SE individual task performance.

| Subtask | W/O Finetuning | Valid. | Random | BM25 | Shapley | Influence Functions | DSIR | RDS | LESS | Full | ROSE (ours) |
|---|---|---|---|---|---|---|---|---|---|---|---|
| academia | 11.0 | 7.0 | 13.5 | 17.5 | 12.0 | 15.0 | 11.5 | 14.0 | 19.5 | 17.5 | 25.5 |
| apple | 7.5 | 7.5 | 12.5 | 15.5 | 11.0 | 12.5 | 14.5 | 15.5 | 19.0 | 21.0 | 25.0 |
| askubuntu | 13.5 | 10.5 | 18.5 | 19.5 | 19.5 | 22.5 | 18.0 | 22.5 | 28.5 | 19.5 | 24.5 |
| english | 6.0 | 5.0 | 8.0 | 15.5 | 14.0 | 13.0 | 17.0 | 13.0 | 16.5 | 11.0 | 24.0 |
| gaming | 9.0 | 8.5 | 11.0 | 12.5 | 11.0 | 11.5 | 12.0 | 11.5 | 13.5 | 12.5 | 18.0 |
| physics | 11.0 | 8.5 | 17.5 | 18.0 | 15.5 | 16.5 | 17.5 | 16.5 | 19.5 | 16.5 | 26.5 |
| security | 10.5 | 13.0 | 17.0 | 17.0 | 18.0 | 14.0 | 17.0 | 17.0 | 24.5 | 20.0 | 28.5 |
| sharepoint | 17.5 | 19.0 | 27.0 | 30.0 | 27.5 | 26.5 | 27.5 | 31.5 | 35.0 | 29.5 | 35.0 |
| softwareengineering | 4.0 | 5.5 | 9.0 | 13.5 | 12.0 | 10.5 | 10.5 | 14.0 | 14.5 | 12.5 | 20.0 |
| workplace | 10.0 | 5.0 | 10.5 | 21.5 | 15.0 | 12.0 | 14.0 | 13.5 | 24.5 | 16.0 | 35.5 |
| Total | 10.0 | 9.0 | 14.3 | 18.0 | 15.6 | 15.4 | 16.0 | 16.9 | 21.5 | 17.6 | 26.2 |

Table 12: HH individual task performance.

| Subtask | W/O Finetuning | Valid. | Random | BM25 | Shapley | Influence Functions | DSIR | RDS | LESS | Full | ROSE (ours) |
|---|---|---|---|---|---|---|---|---|---|---|---|
| harmless-base | 32.5 | 26.6 | 40.6 | 47.9 | 41.5 | 42.6 | 46.5 | 44.7 | 46.5 | 45.7 | 49.9 |
| red-team-attempts | 27.7 | 25.4 | 42.3 | 44.7 | 41.7 | 42.7 | 43.5 | 45.4 | 44.8 | 42.8 | 52.1 |
| Total | 30.1 | 26.0 | 41.7 | 46.3 | 41.6 | 42.6 | 45.0 | 45.1 | 45.6 | 44.2 | 51.0 |

## D    EVALUATION PROMPT

We compare the model's response with the highest-annotated response from the original dataset. For a fair comparison, all models are evaluated on the same prompt using GPT-4-32k-0613. To avoid any bias from GPT towards the order of responses, the model's response is always presented before the response from the dataset.

---

**System Prompt**

Please tell me which response below is more helpful. Do not answer with anything other than Response A or Response B, and do not answer Neither. Your response should use the format:
Comparison: one-sentence comparison and explanation
The better response: Response A or Response B

**User Prompt**

Response A: <Target Model Response>
Response B: <Response with the Highest Score from Original Dataset>

---

## E    ROSE DATA SELECTION ALGORITHM

---

**Algorithm 1** ROSE Data Selection Algorithm

---

1: **Input:** training data corpus $\mathcal{D}$, few-shot SFT validation data $\mathcal{D}_{\text{val}}$, a selection and scoring LLM $\Omega$ and a final training model $\Gamma$.
2: Randomly select 5% data from $\mathcal{D}$ to initially train the selection model $\Omega$ to $\Omega'$.
3: Transform the few-shot SFT validation data $\mathcal{D}_{\text{val}}$ into a few-shot preference validation set $\mathcal{D}'_{\text{val}}$.
4: Backpropagate on selection model $\Omega'$. Extract Adam gradient of each data point in $\mathcal{D}$, and extract SGD gradient of $\mathcal{D}'_{\text{val}}$ based on Equation 6.
5: Compute influence score for each training data point to validation set based on Equation 7.
6: Select the top 5% data with highest influence scores from $\mathcal{D}$ as the selected training set $\mathcal{D}_{\text{train}}$.
7: Train model $\Gamma$ on $\mathcal{D}_{\text{train}}$
8: **Output:** Finetuned instruction LLM $\Gamma'$ for target tasks.

---

## F   ROSE-SELECTED DATA EXAMPLES & CASE STUDIES.

Table 13: Examples with the highest influence scores across various validation datasets. The selected examples for the SHP, SE, and HH datasets are taken from OPEN ASSISTANT 1, COT, and FLAN V2, respectively.

---

**Selected example with the highest influence score to SHP validation set.**

---

<|**user**|>
Se tapo mi inodoro y no se como destaparlo, podrias ayudarme?

<|**assistant**|>
Si tu inodoro no desagua, o lo hace con mucha dificultad, puedes intentar desatascar tu inodoro ejerciendo presión con una fregona enrollada. Usa una fregona vieja que puedas desechar, especialmente si tu inodoro está sucio, y si aún así no lo consigues tal vez deberías llamar a un profesional.

---

**Selected example with the highest influence score to SE validation set.**

---

<|**user**|>
If A congregation is watching the pope give a speech. does that mean that the pope isn't giving a speech? Options:
- yes - it is not possible to tell
- no - Let's solve step-by-step:
The question and answer:

<|**assistant**|>
The Pope cannot simultaneously be giving and not giving a speech at the same time. The answer is no.

---

**Selected example with the highest influence score to HH validation set.**

---

<|**user**|>
Here is a review left by a customer on a product. Would you say he was satisfied or dissatisfied? Title: Lots of Slides to Scan? READ THIS FIRST! Review: While both have the excellent HP software that lets you make color, cropping, light/dark, sharpness adjustments before AND after scanning, the G4010 requires that you hold the lid open with one hand and insert/remove the slides in the slot in the lid with the other! The slot is wide enough for the plastic negatives insert and too wide for slides so they tend to overlap each other. It can be difficult to get them to stay straight. The G4050 model has a template that lays flat on the glass and holds up to 16 slides at a time rather than just 5. And, I found I could not even insert all 5 slides into the lid slot as I could not get them out without slipping and touching the slides. This is not a problem with the G4050 since the template indent for each slide makes removal very easy. And, gravity is working with you, not against you and the lid stays upright so you can use both hands to insert the slides. The G4050 is definitely worth the extra. Answer:

<|**assistant**|>
Dissatisfied

---

Table 14: Comparison of responses generated from different selection methods for the test data examples, including random selection and the response from original dataset.

| A SHP test data example: Question from askbaking subtask |
|---|

**User:** How many hours is över nightfor letting a yeast dough rest in the fridge? I want to make Brioche and it says to let the dough chill in the fridge overnight. Is there a maximum of hours I can let it rest before it starts to overproof? The recipe calls for 30g of fresh yeast to 500g flour if that is relevant.

| Response of random selection | Original response from dataset | Response of ROSE |
|---|---|---|
| **Assistant:** If let to proof for a day, most breads' dough will be over proofed. It isn't recommended to let the dough sit for more than 12 hours in the refrigerator. Heat will cause the dough to rise quickly so keep it at cool temperatures. As 30g of yeast for 500g flour produces quite a good dough, you may want to scale down your recipe. | **Assistant:** I have left my brioche in the fridge for almost 36 hours. The cold in your fridge is just inhibiting the fermentation process, the longer the process, the more flavor development there is. But I try to aim for at least 8-10 hours, but if I can wait 24 hours, that's my ideal. | **Assistant:** It's best to let the dough chill in the fridge overnight. The maximum hours you can let it rest is 16 hours. The maximum hours you can let it rest is 16 hours. The maximum hours you can let it rest is 16 hours. The maximum hours you can let it rest is 16 hours. |

| A SE test data example: Question from academia subtask |
|---|

**User:** A friend of mine had failed core courses during his master's program. He had to retake the exams and passed them with minimum grade. However, his research output was good and he was able to publish a journal paper (first author, IF 4). Now he has been accepted for a PhD in Canada on the same topic as his master's. He will be working with application of the courses he had failed in. Should he continue with his PhD journey? At the university he will be carrying out his research, they don't have to take any coursework. So, he would have to continue with his research without having to take any courses. Is he suitable for PhD? As far I know, PhD students are supposed to be proficient in the coursework.

| Response of random selection | Original response from dataset | Response of ROSE |
|---|---|---|
| **Assistant:** I assume the mentioned research output meant publication record. Have your friend thought of discussing this with his supervisor/adviser about what research problems he find most interesting, and from there leading to shorter term and broader applicable research goals? One way to quickly establish own reputation and related areas of interest are LaTeX documents: - academic publications one or more early publications with his supervisor/adviser / or in other top-tier journals; | **Assistant:** I'm going to answer here as if you are the friend; If the university accepts you into a doctoral program they have fairly high confidence that you will be successful. Otherwise they are just wasting scarce resources, including money, space, and faculty time. It may be that you have some deficiencies and I hope that the educational system in place gives you the time and opportunity to fill any gaps, but you probably also have some compensating points in your favor. | **Assistant:** The fact that he was able to publish a journal paper despite failing core courses during his master's program is a good sign that he is suitable for PhD. It shows that he has the ability to produce high-quality research and can overcome challenges. However, it is important to consider whether he is committed to completing his PhD journey and whether he has the support and resources necessary to succeed. |

| A HH test data example: Question from red-team-attempts subtask |
|---|

**User:** Human: I can't decide what type of coffee I should buy. Do you have any suggestions? Assistant: What are you looking for in a coffee? Human: I'm looking for a coffee that isn't too strong or bold. I also like coffee that has notes of vanilla. Assistant: I'm thinking you might like to try a Vanilla Bean blend, with low-acidity Arabica beans from Ethiopia. Human: Can I purchase those at any grocery store? Assistant: You can, but they're most common in specialty coffee shops that roast their beans on site. Human: Okay, thank you. What would you suggest if I want a coffee that has a really bold flavor? Assistant:

| Response of random selection | Original response from dataset | Response of ROSE |
|---|---|---|
| **Assistant:** For that, you'd want to try a light espresso roast. This type of brew helps bring out the natural flavors of the coffee, and it's also more caffeinated. Just make sure to read the ingredients if you've got a sensitivity to acidity! | **Assistant:** Have you tried a Kona blend, with notes of tropical fruit and natural sweetness? | **Assistant:** If you like a really bold flavor, you might like a dark roast. These are usually made from Robusta beans, which are more bitter and have a higher caffeine content. |

