# OpenReview forum: "ROSE: A Reward-Oriented Data Selection Framework for LLM Task-Specific Instruction Tuning"
_ICLR.cc/2025/Conference — ICLR 2025 Conference Withdrawn Submission_

### Official Review · Reviewer_YK9g · 2024-10-29

**Soundness:** 2
**Presentation:** 3
**Contribution:** 2
**Rating:** 5
**Confidence:** 4

**Summary:**

This paper focuses on the data selection problem for task-specific instruction tuning of Large Language Models (LLMs). It addresses issues with previous works like LESS, which used influence functions for data selection: minimizing validation loss does not monotonically increase performance. The authors propose maximizing reward value on the validation set (minimizing pairwise preference loss) as an objective to replace the validation set loss (next token prediction loss) gradient in LESS. Their experimental results on preference benchmarks show improved effectiveness compared to previous methods. Analysis experiments also partially demonstrate that a decrease in pairwise preference loss correlates more strongly with improved test win rates.

**Strengths:**

1. Building on LESS, this paper conducts valuable exploration into differentiable metrics beyond cross-entropy loss for data selection procedures. It identifies reward value as a potentially more beneficial objective for preference tasks.

2. ROSE's gradient norm cleverly addresses the issue in LESS where sequence length affected the influence function.

**Weaknesses:**

1. ROSE's effectiveness has only been validated on the Preference Benchmark. However, to my knowledge, LESS has shown excellent performance across various task formats such as MMLU, TYDIQA, and BBH. I suspect this limitation is due to the nature of the pairwise preference loss, which may restrict ROSE's ability to extend to other tasks.

2. Given that ROSE introduces pairwise preference loss calculations in the data selection process, I'm unsure whether this increases the method's computational complexity. This includes the asymptotic complexity, wall-clock runtime (measured in single A100 GPU hours), and associated storage costs for different stages such as Warmup LoRA Training, Gradient Features Computation, and Data Selection. If these costs significantly exceed full data training costs, it could potentially diminish the practicality of this method.

**Questions:**

1. As mentioned in Weakness 1, could the authors test ROSE's performance compared to previous methods (LESS) on MMLU, TYDIQA, and BBH?  I believe this would strongly demonstrate ROSE's versatility.

2. Regarding Weakness 2, could the authors compare the computational complexity of ROSE with LESS and full data training? (including  the asymptotic complexity, wall-clock runtime,  and associated storage costs for different stages such as Warmup LoRA Training, Gradient Features Computation, and Data Selection). This would give us a clearer understanding of ROSE's cost.

If the authors address these concerns, I would be inclined to increase my rating.

---

### Official Review · Reviewer_kzgG · 2024-10-30

**Soundness:** 3
**Presentation:** 3
**Contribution:** 3
**Rating:** 5
**Confidence:** 3

**Summary:**

This paper focuses on the data selection problem for task-specific instruction tuning of LLMs. Different from previous methods that primarily rely on the crafted similarity metrics to select training data that aligns with the test data distribution, the proposed method leverages pairwise preference loss as a reward signal to optimize data selection for task-specific instruction tuning.
Specifically, the proposed method adapts an influence formulation to approximate the influence of training data points relative to a few-shot preference validation set to select the most task-related training data points.
Experimental results show the effectiveness of the proposed method.

**Strengths:**

1. This paper attempts to address an important question and proposes an effective method that achieves better performance than the compared methods.
2. The motivation of this paper is clear and the proposed method is sound. The technical approach is sound and well-justified, with a clear connection to the theoretical underpinnings of Direct Preference Optimization (DPO) and influence functions.
3. The paper is well-organized and clearly written. The introduction provides a good motivation for the work, and the related work section is comprehensive. The figures and tables are informative and support the narrative effectively.

**Weaknesses:**

1. Lack of comparison with up-to-date task-specific methods [1,2].

2. Evaluation Benchmarks: This method claims to be task-specific, yet the evaluation datasets used are general open-source
preference benchmarks. Is there a need for further evaluation on specific tasks? For example: summarization.



[1] One Shot Learning as Instruction Data Prospector for Large Language Models

[2] Recost: External knowledge guided data-efficient instruction tuning

**Questions:**

None.

---

### Official Review · Reviewer_U4G4 · 2024-10-30

**Soundness:** 2
**Presentation:** 3
**Contribution:** 2
**Rating:** 5
**Confidence:** 5

**Summary:**

This paper proposes a data selection method named ROSE (Reward-Oriented inStruction data sElection) for task-specific instruction fine-tuning of large language models (LLMs).
ROSE optimizes the data selection for task-specific instruction fine-tuning by using reward signals instead of the traditional loss minimization. This method utilizes the pairwise preference loss as a reward signal, enabling the selected data to better enhance the model's performance in actual tasks.
The experimental results show that the ROSE method can achieve comparable results on multiple benchmark datasets to those obtained using the complete training dataset with only 5% of the training data selected, and it outperforms other advanced data selection methods. This indicates that ROSE can effectively improve the task-specific performance of the model while reducing the training cost. ROSE not only performs excellently on different datasets but also demonstrates its strong generalization ability across various model architectures. The generality of this method makes it potentially valuable in various application scenarios, especially in cases where efficient data selection is required.

**Strengths:**

The author has put forward a new anchor point for large models to screen data. By using reward signals instead of the traditional loss minimization, the data selection for task-specific instruction fine-tuning is optimized. This method utilizes pairwise preference loss as a reward signal, enabling the selected data to better enhance the performance of the model in actual tasks.

**Weaknesses:**

1. Although the ROSE method has achieved remarkable results in data selection, its implementation involves complex gradient calculations and impact estimations, which may lead to high computational costs and implementation complexity, especially when dealing with large-scale datasets and models.
2. The ROSE method relies on a small number of preference validation sets to guide data selection, so the quality of the preference data is crucial to the final selection effect. If the preference data is inaccurate or biased, it may affect the fine-tuning effect of the model. Moreover, there are no relevant experiments in the paper to illustrate that the "optimization direction" of the designed preference validation set and the evaluated dataset for the model is consistent. This makes it impossible for the method in the paper to provide evidence when attempting to demonstrate that the traditional loss minimization is inconsistent with the actual task performance of the model.
3. In addition, most of the methods for screening instruction data of large models in the model comparison are some relatively old baselines. There is a lack of comparison with some more recent methods, and very few recent related works are introduced either.
- From quantity to quality: Boosting LLM performance with self-guided data selection for instruction tuning
- One-shot learning as instruction data prospector for large language models
- What makes good data for alignment? A comprehensive study of automatic data selection in instruction tuning.

**Questions:**

Refer to the section on weakness

---

### Official Review · Reviewer_tsAh · 2024-11-02

**Soundness:** 2
**Presentation:** 3
**Contribution:** 3
**Rating:** 5
**Confidence:** 3

**Summary:**

The paper introduces ROSE, a novel framework for data selection in task-specific instruction tuning of LLMs. ROSE shifts the focus from loss minimization to maximizing a task-specific reward, using pairwise preference loss as a guiding signal for data selection. The experimental results show that ROSE can achieve competitive performance using only 5% of the training data.

**Strengths:**

1.	This study addresses an important issue in instruction tuning for LLMs, focusing on data selection to align model outputs more closely with real-world task performance.
2.	It is novel to focus on reward maximization rather than traditional empirical risk minimization to optimize data selection, which may offer a fresh perspective on enhancing model alignment with human preferences
3.	The experiments are conducted extensively with both qualitative and quantitative evaluations across various model sizes and families.
4.	The Paper is well presented and structured.

**Weaknesses:**

1.	The study uses only 5% of the training dataset for model tuning. It would be beneficial to explore results with other proportions (e.g., 10%, 20%) to understand the method’s effectiveness at varying scales of data selection.
2.	The comparison baseline primarily consists of traditional data selection methods. While ROSE employs GPT-4-32K-0613 model as a judge model, exploring data selection baselines with larger models could further validate ROSE’s effectiveness.
3.	The study uses specific shot numbers (5, 2, and 1) tailored to individual datasets, rather than a generally optimal choice applicable across tasks, which limits insights into the robustness and general applicability.
4.	The use of judge model may impact results significantly. Testing with alternative judge models could help establish the portability and robustness of the approach, and address any potential biases introduced by this specific model choice.

**Questions:**

1.	What is the rationale behind choosing 5% of the training dataset for model tuning and whether there were any computational constraints that limited testing with larger proportions.
2.	For a new task, what strategies or heuristics would the authors recommend for determining an optimal shot number?  The ablation in Section B.1 suggests inconsistencies across datasets in shot number selection, so further guidance on this process could be valuable.
3.	How does the proposed method's computational complexity scale with larger datasets?
4.	See the Weaknesses

---

### Official Review · Reviewer_QMmk · 2024-11-03

**Soundness:** 2
**Presentation:** 2
**Contribution:** 3
**Rating:** 5
**Confidence:** 4

**Summary:**

Instruction tuning aims at improving performance on the target task by selecting training data that aligns with the test dataset distribution. The authors observe that the next-token-prediction loss fails to exhibit a monotonic relationship with down-stream task performance for task specific instruction tuning. They propose leveraging pairwise preference loss as a reward signal to optimize data selection, switching loss minimization to reward maximization. The method approximates the influence of training data points relative to a few-shot preference validation set to select the most task-related training data points. The experimental results demonstrate the efficacy of the proposed method even under the selection of 5% of the training data.

**Strengths:**

1. The method is simple but effective to replace the next-token-prediction gradient in LESS method with the DPO gradient.

2. The experimental results validate the effectiveness of the proposed method, which is impressive to surpass the performance of the full dataset training version.

**Weaknesses:**

1. The authors claim that the validation loss fails to exhibit a monotonic relationship with the target task performance, which is counter-intuitive in machine learning theory. It would be better to provide more support evidence in the introduction section, such as experimental table etc.

2. The relationship between pairwise preference loss and win rate depicted in Figure3 is insufficient to substantiate the claim of "a more consistent correlation between reduced validation loss and increased test win rates".

3. The paper lacks experiments examining the influence of pairwise preference pairs in the validation dataset, which are crucial because the entire framework is grounded in DPO theory.

4. It would enhance the paper to display the performance curve as the number of selected training data increases, ranging from 5% to 100%.

5. The method's reliance on a pairwise preference validation dataset to calculate the influence score for each training sample is burdensome. Moreover, if a preference dataset can be identified or constructed, why not train the model directly on it?

6. If I understand the main point of the paper correctly, the authors suggest that the distribution mismatch between training and test data results in a misalignment between loss and target task performance. To ensure precision and avoid ambiguity, the authors should be more careful with their wording. For instance, the statement "it is widely acknowledged that next-token prediction loss often fails to accurately reflect a model’s real-world performance" should specify that this discrepancy arises due to the violation of the i.i.d. assumption.

**Questions:**

1. In section 3.2.3, why is the L_{ROSE} not affected by the sequence length? The vanilla DPO loss is the summation of the cross-entropy of tokens in the sequence. From what I understand, SimPO normalizes the loss by sequence length, yet the proposed method in the paper continues to use the original DPO loss.

2. The connection between the Influence Estimation Scheme and the ROSE optimization objective seems tenuous. What is the connection between gradient L_{val} and pairwise reward?

---

### Note · Authors · 2024-11-20

**Comment:**

We would like to request the withdrawal of our submission titled "ROSE: A Reward-Oriented Data Selection Framework for LLM Task-Specific Instruction Tuning" (Paper ID: 12889) from the ICLR review process. After careful consideration, we have decided to revise and significantly improve the content of our work before resubmitting it to a future venue.

We deeply appreciate the reviewers' time and constructive feedback, which have provided valuable insights for refining our research. Thank you for understanding.

**Withdrawal Confirmation:**

I have read and agree with the venue's withdrawal policy on behalf of myself and my co-authors.